# VideoTetris:
# Towards Compositional Text-to-Video Generation

**Ye Tian**[1][*]    **Ling Yang**[1][†][*]    **Haotian Yang**[2]    **Yuan Gao**[2]    **Yufan Deng**[1]    **Jingmin Chen**[1]
**Xintao Wang**[2]    **Zhaochen Yu**[1]    **Xin Tao**[2]    **Pengfei Wan**[2]    **Di Zhang**[2]    **Bin Cui**[1][†]
[1]Peking University    [2]Kuaishou Technology
Project: videotetris.github.io    Code: https://github.com/YangLing0818/VideoTetris

## Abstract

Diffusion models have demonstrated great success in text-to-video (T2V) generation. However, existing methods may face challenges when handling complex (long) video generation scenarios that involve multiple objects or dynamic changes in object numbers. To address these limitations, we propose VideoTetris, a novel framework that enables compositional T2V generation. Specifically, we propose *spatio-temporal compositional diffusion* to precisely follow complex textual semantics by manipulating and composing the attention maps of denoising networks spatially and temporally. Moreover, we propose an enhanced video data preprocessing to enhance the training data regarding motion dynamics and prompt understanding, equipped with a new *reference frame attention* mechanism to improve the consistency of auto-regressive video generation. Extensive experiments demonstrate that our VideoTetris achieves impressive qualitative and quantitative results in compositional T2V generation.

## 1  Introduction

With the significant development of diffusion models [1–3] recently, advanced text-to-video models [4–7] have emerged and demonstrated impressive results. However, these models often struggle with generating complex scenes following compositional prompts, such as "*A man on the left walking his dog on the right*", which requires the model to compose various objects spatially and temporally. Moreover, with a growing interest in generating long videos, existing methods [8–10] try to explore multi-prompt long video generation, which is typically limited to simple single-object scene changes. These methods fail to manage scenarios where the number of objects changes dynamically, often resulting in bizarre transformations that do not accurately follow the input text.

To overcome these challenges, we introduce VideoTetris, a novel and effective diffusion-based framework to enable compositional text-to-video generation. Firstly, we define compositional video generation as encompassing two primary tasks: **(i) Video Generation with Compositional Prompts**, which involves integrating objects with various attributes and relationships into a complex and coherent video; and **(ii) Long Video Generation with Progressive Compositional Prompts**, where 'progressive' refers to the continuous changes in the position, quantity, and presence of objects with different attributes and relationships. Then, we introduce a novel **Spatio-Temporal Compositional Diffusion**, which manipulates the cross-attention value of denoising network temporally and spatially, synthesizing videos that faithfully follow complex or progressive instructions. Subsequently, to enhance the ability of long video generation models to grasp complex semantics and generate intricate scenes encompassing various attributes and relationships, we propose an **Enhanced Video**

---

[*]Contributed equally. Contact: yangling0818@163.com
[†]Corresponding Authors.

38th Conference on Neural Information Processing Systems (NeurIPS 2024).

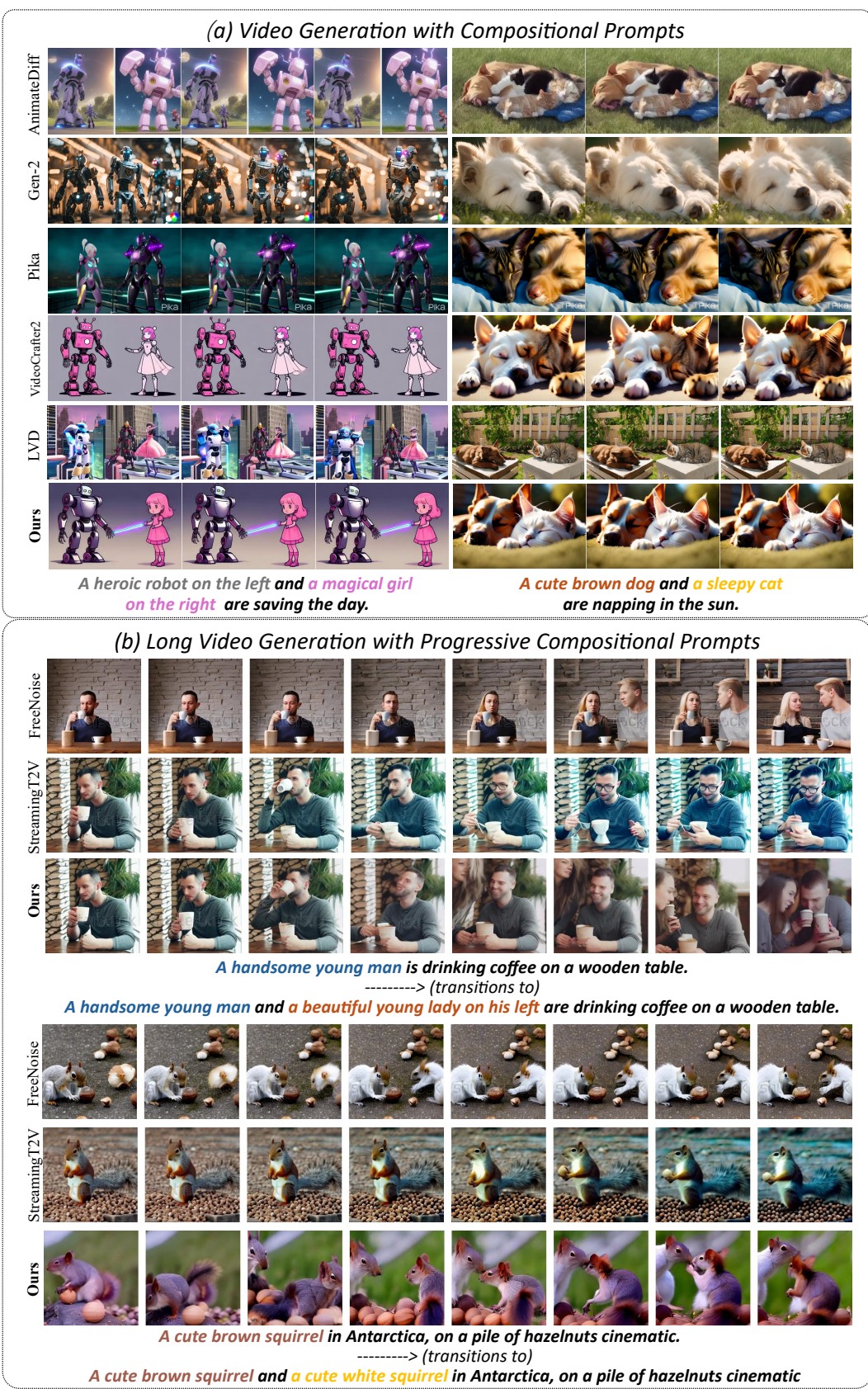

Figure 1: *(a)*: Comparison in Video Generation with Compositoinal Prompts. *(b)*: Comparision in Long Video Generation with Progressive Compositional Prompts. VideoTetris demonstrates superior performance, exhibiting **precise adherence to position information, diverse attributes, interaction, consistent scene transitions, and high motion dynamics** in compositional video generation.

**Data Preprocessing** pipeline, augmenting the training data with enhanced *motion dynamics* and *prompt semantics*, enabling the model to perform more effectively in long video generation with progressive compositional generation. Finally, we propose a consistency regularization method, namely **Reference Frame Attention**, that maintains content consistency in a coherent representation space with latent noise while being capable of accepting arbitrary image inputs, ensuring the consistency of multiple objects across different frames and positions. fig. 1(a) showcases our VideoTetris's superior performance in compositional short video generation. We accurately compose two distinct objects with their own attributes while maintaining their respective "left" and "right" positions and ensuring natural interaction between multiple objects. As for long video generation comparisons in fig. 1(b), FreeNoise [10] either depicts characters appearing abruptly and inexplicably transforming a man into a woman or depicts a squirrel transforming from a hazelnut. StreamingT2V [11] fails to incorporate information about new characters altogether, ignoring quantity information and exhibits severe color distortion in later stages. In contrast, our VideoTetris excels in generating long videos with progressive compositional prompts, seamlessly integrating new characters into the video scenes while maintaining consistent and accurate positional and quantity information. Notably, in generating long videos of the same length, FreeNoise produces only minor variations within the same scene, whereas VideoTetris demonstrates significantly higher motion dynamics, resulting in outputs that more closely resemble long narrative videos.

Our contributions are summarized as follows: **1**). We introduce a Spatio-Temporal Compositional Diffusion method for handling scenes with multiple objects and following progressive complex prompts. **2**). We develop an Enhanced Video Data Preprocessing pipeline to enhance auto-regressive long video generation through motion dynamics and prompt semantics **3**). We propose a consistency regularization method with Reference Frame Attention that maintains content coherence in compositional video generation. **4**). Extensive experiments show that VideoTetris can generate state-of-the-art quality compositional videos, as well as produce high-quality long videos that align with progressive compositional prompts while maintaining the best consistency.

## 2   Related Work

**Text-to-Video Diffusion Models**   The field of text-to-video generation has seen significant advancements with the progress of diffusion models [1, 12, 13] and the development of large-scale video-text paired datasets [14, 15]. Early works such as LVDM [16] and ModelScope [7], adapted 2D image diffusion models by flattening the U-Net architecture to a 3D U-Net and training on extensive video datasets. Subsequently, methods like AnimatedDiff [5] have incorporated temporal attention modules into the existing 2D latent diffusion models, preserving the established efficacy of T2I models. More recently, several transformers-based diffusion methods [17, 18, 6] have enabled large-scale joint training of videos and images, leading to significant improvements in generation quality.

**Long Video Generation**   Most existing text-to-video diffusion models have been trained on fixed-size video datasets due to the increased computational complexity and resource constraints. Consequently, these models are often limited to generating a relatively small number of frames, leading to significant degradation in quality when tasked with generating longer videos. Several advancements [19, 8, 10, 20] have sought to overcome this limitation through various strategies. More recently, Vlogger [9] and SparseCtrl [21] employ a masked diffusion model for conditional frame input. Although these masked diffusion approaches facilitate longer video generation, they often encounter model inconsistencies and quality degradation due to domain shifts in input. StreamingT2V [11] proposes a new paradigm, utilizing a ControlNet [22]-like conditioning scheme to enable auto-regressive video generation. However, due to the low quality of the training data, the final video outputs often exhibit inconsistent and low-quality artifacts.

**Compositional Video Generation**   While current video generation models can synthesize text-guided videos, they often face challenges in generating videos featuring multiple objects or adhering to multiple complex instructions, which requires the model to compose objects with diverse temporal and spatial relationships. In the realm of text-to-video diffusion models, exploration of such scenarios remains incomplete. Several text-to-image methods like RPG [23] leverage additional layout or regional information to facilitate more intricate image generation [24–27, 23]. Within video diffusion techniques, approaches like LVD [28] and VideoDirectorGPT [29] employ a layout-to-video generator to produce videos based on spatial configurations. However, these layout-based methods often offer

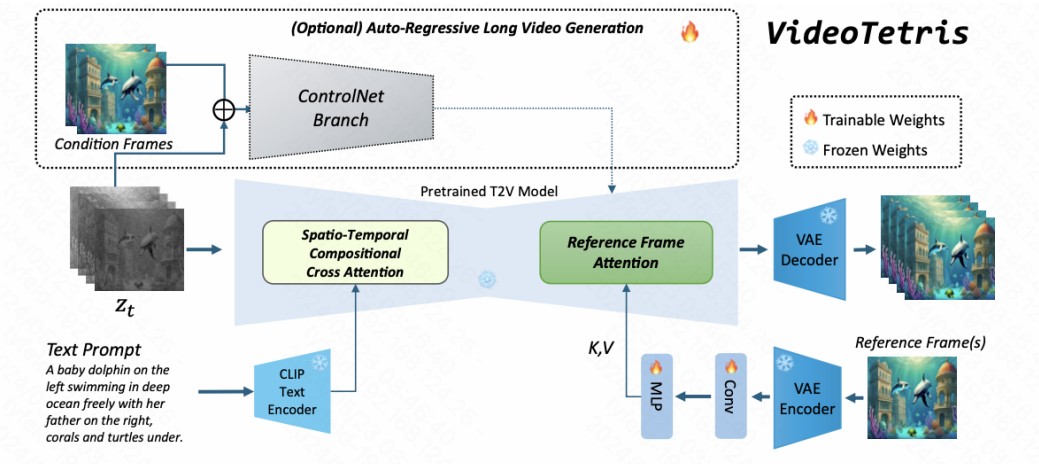

Figure 2: The overall pipeline of VideoTetris. We introduce Spatio-Temporal Compositional module for compositional video generation and Reference Frame Attention for consistency regularization. For longer video generation, a ControlNet [22]-like branch can be adopted for auto-regressive generation.

only rudimentary and suboptimal spatial guidance, struggling particularly with overlapping objects, thereby resulting in videos with unnatural content. In contrast, our method adopts a compositional region diffusion approach. By explicitly modeling the spatial positions of objects with cross attention maps, our approach allows the objects to naturally integrate and blend during the denoising process, resulting in more realistic and coherent video output.

## 3 Method

**Overview**  In this section, we introduce our method VideoTetris for compositional text-to-video generation. Our goal is to develop an efficient approach that enables text-to-video models to handle scenes with multiple objects and follow sequential complex instructions. We first introduce Spatio-Temporal Compositional Region Diffusion in section 3.1, which allows different objects to naturally integrate and blend during the denoising process in a training-free manner. Furthermore, for the task of generating long videos with progressive complex prompts, we construct an auto-regressive model based on the ControlNet [22] architecture and introduced a Enhanced Video Data Preprocessing pipeline in section 3.2 to collect a high-quality video-text pair dataset to train our auto-regressive model for enhanced motion dynamics and prompt understanding. Combined with Spatio-Temporal Compositional Region Diffusion, our auto-regressive model can generate long videos with seamless transitions between diverse target scenes. Finally, we propose a consistency regularization with Reference Frame Attention in section 3.3 for better object appearance preserving.

### 3.1 Spatio-Temporal Compositional Diffusion

**Motivation**  To achieve natural compositional generation, a straightforward approach is to use the layout as a condition to guide the generation process. However, this method presents several challenges: (i) Requiring large-scale training. Given the significant potential for improvement in layout-to-image models, training a layout-to-video model or training temporal convolution and attention layers for a layout-to-image model would require substantial computational resources and may struggle to keep pace with the latest advancements in text-to-video models. (ii) Layout-based generation models impose significant constraints on object bounding boxes. For long video duration, the need to continuously adjust the positions and sizes of these boxes to maintain coherent video content introduces a complex planning process, which adds complexity to the overall method. Therefore, instead of training a layout-to-video model, we utilize cross-attention for precise generation [30–35] and propose a training-free approach that directly adjusts the cross-attention of different targets [23, 36–39], as is shown in fig. 3. This approach aims to overcome the limitations of layout-based methods and leverage the potential of more flexible and efficient generation techniques.

**Localizing Subobjects with Prompt Decomposition**  For a given prompt $p$, we first decompose it temporally into contents at different frames: $p = \{p^1, p^2, \cdots, p^t\}$, where $t$ denotes the total number of frames and $p^i$ denotes the given text prompt at $i$-th frame. Subsequently, for the $i$-th

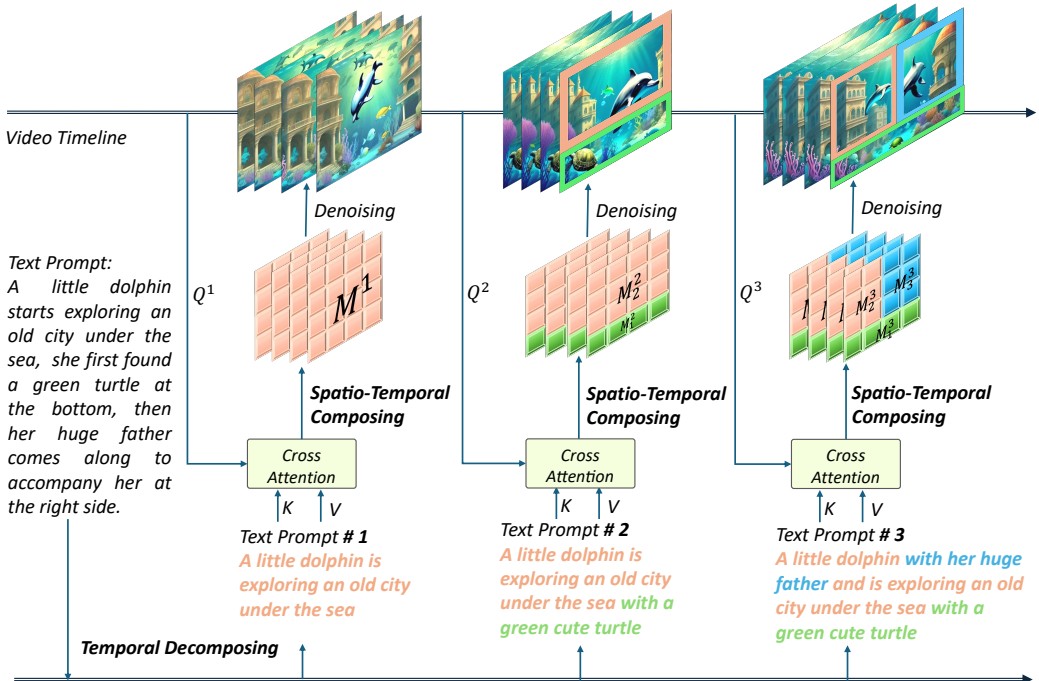

Figure 3: Illustration of Spatio-Temporal Compositional Diffusion. For a given story "A little dolphin starts exploring an old city under the sea, she first found a green turtle at the bottom, then her huge father comes along to accompany her at the right side.", we first decompose it temporally to Text Prompt #1, #2 and #3, then we decompose each of them spatially to compute each sub-region's cross attention maps. Finally, we compose them spatio-temporally to form a natural story.

frame, we decompose the original $p^i$ spatially into different sub-objects: $\{p_0^i, p_1^i, \cdots, p_n^i\}$ with their corresponding region masks $M^i = \{M_0^i, M_1^i, \cdots, M_n^i\}$, where $n$ denotes the number of different objects. In this way, we decompose a prompt list temporally and spatially to acquire each sub-object's corresponding region information in the video timeline. We then calculate the cross attention value for the $j$-th sub-object at $i$-th frame as follows:

$$\text{CrossAttn}_j^i = \text{Softmax}\left(\frac{Q^i(K_j^i)^T}{\sqrt{d}}\right)V_j^i \odot M_j^i, \quad K = W_K * \phi(p_j^i), V = W_V * \phi(p_j^i) \tag{1}$$

where $Q^i$ represents the query for the latent frame features, $W_K, W_V$ are linear projections, $\phi$ denotes the text encoder, and $d$ is the latent projection dimension of the latent frame features.

**LLM-based Automatic Spatio-Temporal Decomposer (Optional)** Alternatively, the spatio-temporal decomposition process can directly utilize a Large Language Model (LLM) to automate tasks, given the robust performance of LLMs in language comprehension, reasoning, summarization and region generation abilities [23, 28, 27, 26]. We employ the in-context learning (ICL) capability of LLMs and guide the model to use chain-of-thought (CoT) [40] reasoning. Concretely, we first guide the LLM to decompose the story temporally, generating frame-wise prompts, and reception each one of them with LLM for better semantic richness. Then we use another LLM to decompose each prompt spatially into multiple prompts corresponding to different objects, assigning a region mask to each sub-prompt. The specific prompt templates that include task rules (instructions), in-context examples (demonstrations) are detailed in table 4, table 5 and table 6 of appendix A.1.

**Spatio-Temporal Subobjects Composition** After we decompose the original prompt list temporally and spatially, we then compose them together from spatial to temporal. To this end, we first compute the cross-attention value of all sub-objects $\text{CrossAttn}_{region}^i$ at $i$-th frame with:

$$\text{CrossAttn}_{region}^i = \sum_{j=0}^{n} \text{CrossAttn}_j^i \tag{2}$$

Subsequently, to ensure a cohesive transition across the boundaries of distinct regions and a seamless integration between the background and the entities within each region, we employ the weighted sum

of the CrossAttn$_{region}$ and the CrossAttn$_{original}$ for the original compositional prompt $p$ with :

$$\text{CrossAttn}^i_{original} = \text{Softmax}(\frac{Q^i(K^i)^T}{\sqrt{d}})V^i, \quad K = W_K * \phi(p^i), V = W_V * \phi(p^i) \tag{3}$$

$$\text{CrossAttn}^i = \alpha * \text{CrossAttn}^i_{original} + (1 - \alpha) * \text{CrossAttn}^i_{region}.$$

Here $\alpha$ parameter is utilized to adjust the balance between global information and individual characteristics, aiming to achieve video content more aligned with human aesthetic perception. Finally, we naturally concatenate all the cross-attention values computed along the temporal dimension:

$$\text{CrossAttn} = \text{Concat}(\text{CrossAttn}^1, \text{CrossAttn}^2, \cdots, \text{CrossAttn}^t) \tag{4}$$

In this way, either for a pre-trained text-to-video model such as Modelscope [7], Animatediff [5], VideoCrafter2 [4] and Latte [6], or an auto-regressive model for longer video generation like StreamingT2V[11], this approach can be directly applied in a training-free manner to obtain compositional, consistent and aesthetically pleasing results.

## 3.2 Enhanced Video Data Preprocessing

**Enhancement of Motion Dynamics**  For auto-regressive video generation, we empirically find StreamingT2V [11] is the most effective in producing consistent content. However, there is a notable tendency for the occurrence of poor-quality cases and color degradation in the later stages of video generation. We attribute this issue to the suboptimal quality of the original training data. To enhance the motion consistency and stability of long video generation, it is imperative to filter the video data to retain high-quality content with consistent motion dynamics. Inspired by Stable Video Diffusion [41], we empirically observed a significant correlation between a video's optical flow [42] score its motion magnitude. Excessively low optical flow often corresponds to static video frames, while excessively high optical flow typically indicates frames with intense changes. To ensure the generation of smooth and suitable video data, we filter Panda-70M [15] by selecting videos with average optical flow scores computed by RAFT [43] falling within a specified range ($s_1$ to $s_2$).

**Enhancedment of Prompt Semantics**  While the Panda-70M's videos exhibit the best visual quality, the paired prompts tend to be relatively brief, which conflicts with our objective of generating videos that adhere to intricate, detailed, and compositional prompts. Directly using such data for training can result in a video generation model that inadequately comprehends complex compositional prompts. Inspired by recent text-to-image research [23, 44, 45], it has been demonstrated that high-quality prompts significantly enhance the output quality of visual content. Therefore, after filtering the initial set of videos, we perform a recaptioning process on the selected samples to ensure they are better aligned with our objectives. We employ three multimodal LLMs to generate spatio-temporally intricate and detailed descriptions of each video, followed by a local LLM to consolidate these descriptions, extract common elements, and add further information. More details on this process can be found in appendix A.2.

## 3.3 Consistency Regularization with Reference Frame Attention

Given our approach involves the addition and removal of different objects in long videos, maintaining the consistency of each object throughout the video is crucial for final outputs. Most consistent ID control methods, such as IP-Adapter [46], StreamingT2V [11], InstantID [47], and Vlogger [9], typically encode reference images using an image encoder, often CLIP [48], and then integrate the results into the cross-attention block. However, since CLIP is pre-trained on image-text pairs, its image embeddings are designed to align with text. Consistency control, on the other hand, focuses on ensuring that the feature information of the same object in different frames is similar, which does not involve text. We hypothesize that using CLIP for this purpose is an indirect approach and propose Reference Frame Attention to maintain the inter-frame consistency of object features.

Formally, we first directly encode the reference images, which are usually the initial frames where the object appears, using the same autoencoder as the pre-trained T2V model. This ensures that the computational target during latent denoising is spatially consistent with the reference target within the hidden representation space. We then train a 2D convolutional layer and projection layer that are structurally identical to those in the original pipeline. This process can be represented as:

$$x_{ref} = W(\text{Conv}(\text{AutoEncoder}(f_{k:k+l}))), \tag{5}$$

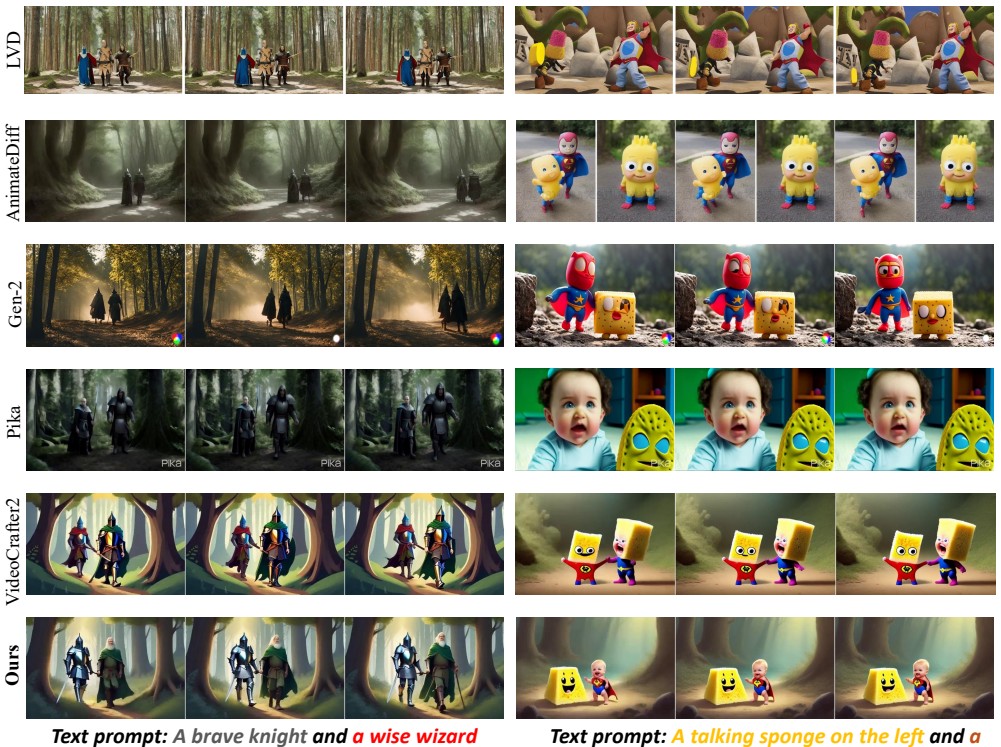

**Text prompt: *A brave knight* and *a wise wizard* are journeying through a forest.**

**Text prompt: *A talking sponge on the left* and *a superhero baby* on the right are having an adventure.**

Figure 4: Qualitative Results of Video Generation with Compositional Prompts in Comparision with SOTA Text-to-Video Models

where $W$, Conv denote the projection layer and the 2D convolutional layer, $f_{k:k+l}$ denotes the $l$ frames from index $k$ that are chosen for refernce. After encoding, we insert a Reference Frame Attention block in each attention block that calculates the cross-attention between the current object and the reference object, supplementing the existing attention blocks:

$$\text{RefAttn} = \text{Softmax}(\frac{QK^T}{\sqrt{d}})V, \quad K = W_K * x_{ref}, V = W_V * xref \quad (6)$$

It is noteworthy that to ensure the consistency of different objects across various regions, we need to separately multiply the corresponding object's region mask with $Q, K$, and $V$ during this computation process, and in practical applications, when a new object emerges in the auto-regressive long video, we precompute its corresponding $x_{ref}$ in the relevant regions for further process.

## 4 Experiments

### 4.1 Experimental Setups

We conducted our experiments in two specific scenarios: Video Generation with Compositional Prompts and Long Video Generation for progressive Compositional Prompts. For the first scenario, we directly applied our Spatio-Temporal Compositional Diffusion on VideoCrafter2 [4] to generate videos with $F = 16$ frames. For the second scenario, we employed the core ControlNet [22]-like branch from StreamingT2V [11] as the backbone and processed the Panda-70M [15] dataset using the Enhanced Video Data Preprocessing methods in section 3.2 as the training set. For both scenarios, we used ChatGPT[3] to generate 100 different prompts/prompt lists as input to the models, generated 6 videos for each prompt, and randomly selected one for comparison. Additional model hyperparameters and implementation details of VideoTetris are provided in appendix A.5.

---

[3]chat.openai.com

Table 1: Quantitative Results of Video Generation with Compositional Prompts

| Method | VBLIP-VQA | VUnidet | CLIP-SIM |
|---|---|---|---|
| Animatediff [5] | 0.3834 | 0.1921 | 0.8676 |
| VideoCrafter2 [4] | 0.4510 | 0.1719 | 0.9249 |
| Gen-2 [53] (Commercial) | 0.4427 | 0.1503 | 0.9421 |
| Pika [54] (Commercial) | 0.4219 | 0.1782 | **0.9736** |
| LVD [28] | 0.4820 | 0.1934 | 0.8873 |
| VideoTetris (Ours) | **0.5563** | **0.2350** | 0.9312 |

## 4.2 Metrics

To evaluate compositinal video generation, existing metrics, such as CLIPScore [48] and Fréchet Video Distance (FVD) [49], assess coarse text-video and video-video similarity but do not capture detailed correspondences in object-level attributes and spatial relationships. Instead, we extended the T2I-CompBench [50] to the video domain and introduced the following metrics for **compositional text-to-video evaluation**: **VBLIP-VQA**: the average BLIP [51]-VQA score averaged across all frames and **VUnidet**: the average Unidet [52] score averaged across all frames. In addition, we followed previous work [10] and used **CLIP-SIM** [48] to measure the content consistency of generated videos by calculating the CLIP [48] similarity among adjacent frames of generated videos.

## 4.3 Video Generation with Compositional Prompts

**Qualitative Results**  We compare our VideoTetris with several state-of-the-art text-to-video (T2V) models on their ability to generate videos based on complex compositional prompts. These models include open-source options like LVD [28], VideoCrafter2 [4], and Animatediff [5], as well as commercial models Gen-2 [53] and Pika [54]. Using VideoCrafter2 as a backbone, we directly evaluate our Spatio-Temporal Compositional Diffusion module's training-free performance. In fig. 4, we show text-to-video synthesis results. For the prompt, "A brave knight and a wise wizard are journeying through a forest," most models generate two similar characters, blending features and losing individual distinctions. This highlights challenges in semantic alignment and compositional modeling for open-source models. In contrast, our VideoTetris preserves the distinct characteristics of each object and integrates them seamlessly with the background without confining them to fixed regions. For the prompt, "A talking sponge on the left and a superhero baby on the right are having an adventure," models like AnimateDiff split the image, while Runaway Gen-2, Pika, and VideoCrafter2 produce misaligned characters. LVD produces entangled features, resulting in disordered representations. In contrast, our method accurately aligns objects to their specified positions while maintaining high video quality, outperforming other methods. Additional examples in fig. 8 demonstrate our model's capability to handle more complex prompts with multiple objects, maintaining high quality and adherence to compositional semantics.

**Quantitative Results**  We report our quantitative results in table 1. Our VideoTetris achieves the best VBLIP-VQA and VUnidet scores across all models, demonstrating our superiority for complex compositional generation. We also achieved a CLIP-SIM higher than the original backbone VideoCratfer 2[4] and comparable to commercial models thanks to accurate semantic understanding. This proves that better text-video alignment can benefit overall consistency.

**User Study**  For further evaluation, we conducted a user study comparing our method with other video generation models, reported in appendix A.4. Using GPT-4, we collected 100 compositional prompts and generated 100 video samples across diverse scenes, styles, and objects. Users compared model pairs by selecting their preferred video from three options: method 1, method 2, and comparable results.

## 4.4 Long Video Generation for Progressive Compositional Prompts

**Qualitative Results**  We compared our VideoTetris with state-of-the-art long video generation models FreeNoise [10] and StreamingT2V [11]. FreeNoise inherently supports multi-prompts, and we provide StreamingT2V with different prompts at various frame indexes for multi-prompt video generation. We present our qualitative experimental results in fig. 1 and fig. 5. For the multi-

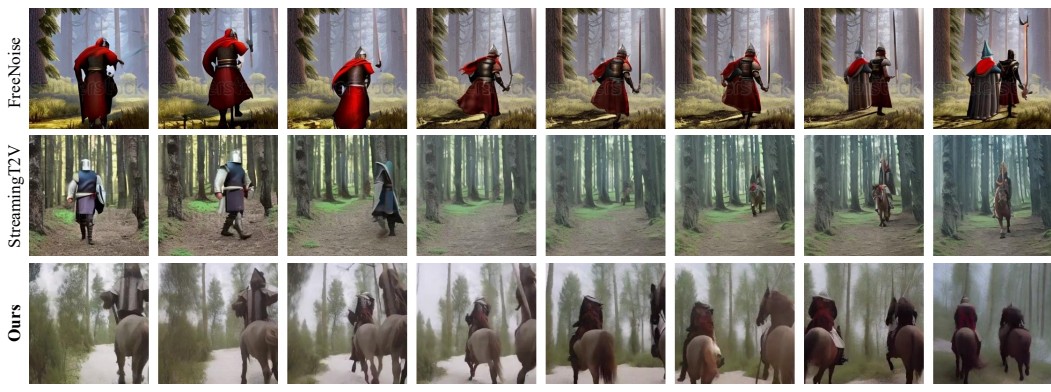

**Progressive Prompts :**
*A brave young knight is journeying through a forest*
*---------> (transitions to)*
*A brave young knight and a wise wizard are journeying through a forest*

Figure 5: Qualitative Results of Long Video Generation for Progressive Compositional Prompts.

prompt sequence from "A brave young knight is journeying through a forest" to "A brave young knight and a wise wizard are journeying through a forest," FreeNoise generates consistent content; however, the new character appears abruptly, and the two characters switch identities by the end. Additionally, FreeNoise consistently produces near-static global motion, with neither the background nor character positions changing. Conversely, StreamingT2V produces bizarre videos in which the knight disappears for half the duration and a merged character appears. In contrast, our method successfully models stable and consistent changes in long videos. The new character appears naturally and integrates seamlessly with the existing background throughout the video. Moreover, our approach achieves significantly more dynamic motion compared to FreeNoise. This further demonstrates our method's capability of generating long videos that fully adhere to the evolving semantics while maintaining overall consistency.

**Quantitative Results** We report our quantitative results in table 2. We achieve the best VBLIP-VQA and VUnidet scores across all models, demonstrating the robust generation capability of our model in compositional video generation. FreeNoise archives a better CLIP-SIM score due to its unique noise scheduling method, but this empirically damages transitions in a natural story.

Table 2: Quantitative Results of Long Video Generation for Progressive Compositional Prompts

| Method | VBLIP-VQA | VUnidet | CLIP-SIM |
|---|---|---|---|
| FreeNoise [10] | 0.4372 | 0.1823 | **0.9706** |
| StreamingT2V [11] | 0.2412 | 0.1367 | 0.6720 |
| VideoTetris (Ours) | **0.4839** | **0.2137** | 0.9521 |

## 4.5 Ablation Study

**Effect of Enhanced Video Data Preprocessing** We conducted an ablation study about the Enhanced Video Data Preprocessing pipeline, and show the results in fig. 6 and table 3. We directly compare our auto-regressive generation results with the original StreamingT2V [11] using the original prompts and comparison methods. fig. 6 demonstrates the significant improvements we achieved. For the given prompt our model better captures the semantics of "early morning sunlight." In addition, we generate long videos with all test prompts in StreamingT2V, and report our MAWE [11], CLIP(image-text alignment), AE [55] and CLIP-SIM scores, which further proves our effectiveness.

Table 3: Quantitative Comparison of Ablation Study.

| Method | MAWE ↓ | CLIP ↑ | AE ↑ | CLIP-SIM ↑ |
|---|---|---|---|---|
| FreeNoise [10] | 49.53 | 32.14 | 4.79 | 0.91 |
| StreamingT2V [11] | 10.26 | 31.30 | 5.07 | 0.93 |
| VideoTetris w/o Reference Frame Attention | 10.21 | 33.50 | 7.21 | 0.92 |
| **VideoTetris (*Ours*)** | **9.98** | **34.80** | **8.07** | **0.96** |

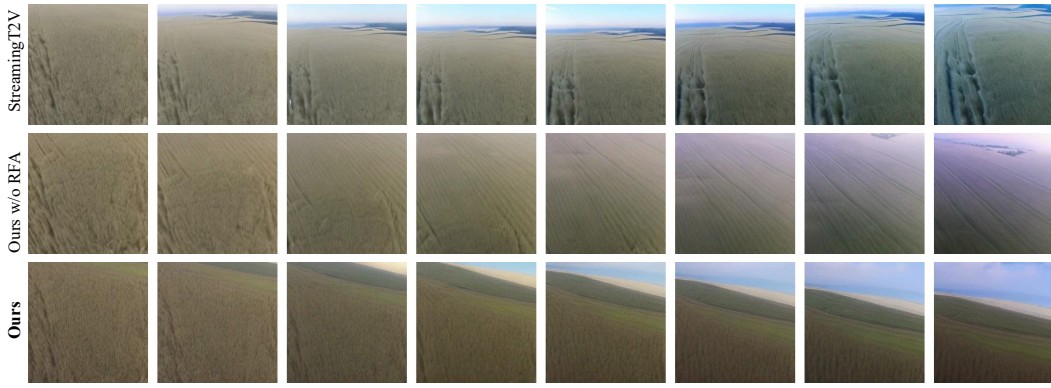

**Text prompt:**
*Close flyover over a large wheat field in the early morning sunlight.*

Figure 6: Ablation Study. Comparison Between the original StreamingT2V [11], VideoTetris w/o Reference Frame Attention and VideoTetris.

**Effect of Reference Frame Attention**  We also conducted an ablation study about the Reference Frame Attention, as demonstrated in fig. 6 and table 3. We observe from the result that our Reference Frame Attention achieves more consistent outputs, and the frequency of color artifacts significantly decreases, resulting in a more uniform overall color. This highlights the benefit of aligning reference and noise information semantically in the latent space. We provide more ablation studies about compositional approaches in appendix A.3.

## 5   Conclusion and Discussion

**Conclusion**  In this study, we addressed the limitations of current video generation models incapable of generating compositional video content and introduced a novel VideoTetris framework that enables high-quality compositional generation. We propose an efficient Spatio-Temporal Compositional module that decomposes and composes semantic information temporally and spatially in the cross-attention space. Additionally, to further enhance consistency in auto-regressive long video generation, we introduced an Enhanced Video Data Preprocessing pipeline and designed a brand new Reference Frame Attention module. Extensive experiment results confirmed the superiority of our paradigm in extending the generative capabilities of video diffusion models.

**Limitations**  Our proposed method can generate both short and long compositional videos. For fixed text-to-video generation, we can directly enhance the compositional performance of existing models. However, for long videos, due to the current performance limitations of long video generation models, our method inevitably encounters some bad cases. Additionally, using ControlNet [22] for auto-regressive long video generation results in huge computation cost and overly strong control information, leading to an excessive number of transition frames.

**Broader Impact**  Recent notable progress in text-to-video diffusion models has opened up new possibilities in creative design, autonomous media, and other fields. However, the dual-use nature of this technology raises concerns about its societal impact. There is a significant risk of misuse of video diffusion models, particularly in the impersonation of individuals. It is essential to emphasize that our algorithm is designed to enhance the quality of video generation, and we do not support or provide means for malicious uses.

## Acknowledgement

This work is supported by National Natural Science Foundation of China (U23B2048, U22B2037), Beijing Municipal Science and Technology Project (Z231100010323002), research grant No. SH-2024JK29 and High-performance Computing Platform of Peking University.

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

# A   Appendix

## A.1   LLM-based Automatic Spatio-Temporal Decomposer

Large Language Models (LLMs) have showcased impressive language comprehension, reasoning, and summarization abilities. Using LLMs for specific region generation has been proven efficient and effective in previous works [23, 28, 27, 26]. In our work, we employ the in-context learning (ICL) capability of LLMs to generate reasonable and natural temporal information and spatial regions, eliminating the need for manual spatio-temporal prompt decomposing for each prompt. We construct prompt templates that include task rules (instructions), in-context examples (demonstrations), and the user's input prompt (test). The specific prompts used for decomposing prompts spatio-temporally and recaptioning sub-prompts are detailed in table 4, table 5 and table 6. In designing our LLM prompts, we focused on clearly defining the roles and guidelines for the LLM, building on insights from previous research [26, 23, 28]. Moreover, by guiding the model to use chain-of-thought(CoT) [40] reasoning, where it articulates its reasoning process during generation, we empirically achieved better outcomes. This CoT method produced more accurate suggestions compared to when the model's reasoning process was not explicitly detailed. In our experiment, we choose GPT-4 [56] as our LLM.

```
# Your Role: Excellent Story Planner

## Objective: Analyze your input descriptions and plan a reasonable story
    by providing the frame-specific prompt.

## Process Steps
1. Read the user input story with his given total frames.
2. Analyze the story, and specify every object and its attribute.
3. Crafting a video timeline with a prompt and its corresponding frame
    index, Keep the frame index an integer multiple of 8.
4. Explain your understanding (reasoning) and then format your result as
    examples.

## Examples

- Example 1
    User prompt: I would like to create a story about a man in a cafe. He
        first drinks coffee alone on a wooden table, and then a young lady
        with blonde hair comes to company. They started chatting joyfully
        at the end. Total frames: 80
    Reasoning: This story contains three main objects: a man, a wooden
        table, and a young lady with blonde hair. We can split the 80
        frames into 3 different parts to construct a story.
    Output: ['0': "A man is drinking coffee on a wooden table", "32": "A
        man and a young lady with blonde hair are drinking coffee on a
        wooden table", "64": "A man and a young lady with blonde hair are
        drinking coffee and chatting joyfully on a wooden table"]
- Example 2:
    ......

Your Current Task: Follow the steps closely and accurately identify frame
    index specific sub-prompts based on the given story and total frames.
    Ensure adherence to the above output format.

User prompt: {the input user prompt}
Reasoning:
```

Table 4: Our full prompt for Decomposing Prompts Temporally.

```
# Your Role: Excellent Prompt Recaptioner
You are an excellent recaptioning bot. Your task is to recaption each
    given prompt with a more descriptive prompt while maintaining the
    original meaning with at least 40 words.
You will be given with an caption of a video, this caption is very short
    and simple, only containing the main entity and perhaps the simplest
    description of the background.
Please take the provided caption and expand to at least 40 words upon it
    by providing additional details. You can start this procedure by
    following these rules:

## Objective: Recaption each given prompt with a more descriptive prompt
    while maintaining the original meaning with at least 40 words.

## Process Steps
1. If the original prompt contains words about the camera view, such as
    "top view of" or "camera clockwise", remember to also contain them in
    the recaption.
2. Describe each entity appearing in this original caption with at least
    more than two adjectives, making every entity as detailed as possible.
3. Using your knowledge to fulfill the background or any other thing that
    should or should not appear in this frames. Adding as much details as
    you can to enrich the caption, but you shouldn't change the original
    meaning of the prompt or any main entity.
4. Your recaption should contain at least 40 words, and you should keep it
    within 60 words.
5. Your answer should strictly follow the form : "Recaption: "
6. Your answer must not contain words like "video" or "frame". Only enrich
    the given prompt.

## Examples

- Example 1:
Original Caption: a man and woman are walking down a hallway
Recaption: a man and woman in business attire are seen walking down a
    hallway in a professional building, engaged in a serious discussion
    with the man holding a book and the woman holding a clipboard,
    reflecting a professional or academic setting.

- Example 2:
    ......

Your Current Task: You will be given a caption of a video, this caption is
    very short and simple, only containing the main entity and perhaps the
    simplest description of the background. Please take the provided
    caption and expand it to at least 40 words by providing additional
    details.

Original Caption: {the input user prompt}
Recaption:
```

Table 5: Our full prompt for Prompt Recaptioning.

```
# Your Role: Excellent Region Planner

## Objective: Analyze your input prompts and plan every object's
    reasonable region in the frame with bounding boxes.

## Process Steps
1. Analyze the given multi-object prompt, consider a reasonable layout.
2. Define square images with top-left at [0, 0] and bottom-right at [1,
    1], and the output Box Format: [Top-left x, Top-left y, Width, Height]
3. Assign each sub-object to a specific region. You can start by splitting
    the original image square.
4. The corresponding regions do not need to be very specific as long as
    the region includes the sub-object and all regions never overlap
5. Output the result, and present every object and its region with a
    bounding box.
## Examples

- Example 1
    User prompt: A handsome young man and a lady with blonde hair are
        drinking coffee on a wooden table.
    Output: ["a handsome young man": "[0.5, 0, 0.5, 0.8]", "a lady with
        blonde hair": "[0, 0, 0.5, 0.8]", "a wooden table": "[0, 0.8, 1,
        0.2]"]
- Example 2:
    ......

Your Current Task: Follow the steps closely and accurately output each
    sub-object's bounding box. Ensure adherence to the above output format.

User prompt: {the input user prompt}
Output:
```

Table 6: Our full prompt for the *LLM Spatial Decomposer*

## A.2 Dataset Prompt Recaptioning

In this section, we detailed our dataset prompt recaptioning process. We first select the top three caption models ranked highest in [57], namely Video-LLaMA [58], Video-ChatGPT [59] and Video-llava [60], and have them generate captions of 40-50 words for each filtered video. We assume that different caption models may be suitable for different types of video input, so we collect outputs from various models to ensure a comprehensive effect. We then append these captions to the original prompt caption provided by Panda-70M. All collected prompts are fed into a local LLM (in our experiment, LLama-3[4]), to consolidate the captions, extract common elements, and add details. The final unified caption, around 40-50 words in length, is used for training each filtered video.

## A.3 Ablations about Effect of Spatio-Temporal Compositionl Diffusion

In this section, we provided detailed explanations and ablations compared with similar prompt decomposing and object composing diffusion methods, LVD [28], Training-Free Layout Control with Cross-Attention Guidance [61], and VideoDirectorGPT [29] in table 7.

The decomposition methods in [28] and [29] only **isolate specific tokens** from the original prompt. This approach struggles with complex attributes or multiple identical objects, making it difficult for the video generation model to understand numeracy and attribute binding, leading to significant performance degradation in these scenarios. In contrast, our decomposition method extracts subobjects and then uses global information for recaptioning, resulting in richer descriptions. Our generated frames are more natural, detailed, and semantically accurate.

---

[4]https://llama.meta.com/llama3/

Next, we have conducted ablation studies about the composition method from [61] and reported the results in the table below. The backward guidance approach in [61] tends to restrict objects within specified boxes, offering low flexibility and poor responsiveness to multiple or overlapping objects. In contrast, our model's local-global information fusion ensures that the final generated images are more harmonious and visually appealing, performing better in compositional generation, even in overlapping regions.

Table 7: Ablation Studies for Spatio-Temporal Compositional Diffusion

| Method | VBLIP-VQA | VUnidet | CLIP-SIM |
|---|---|---|---|
| Ours w/ Decomposing in LVD [28] | 0.5203 | 0.2139 | 0.9303 |
| Ours w/ Decomposing in [29] | 0.4982 | 0.2237 | 0.9178 |
| Ours w/ Composing in VideoDirectorGPT [61] | 0.5112 | 0.1857 | 0.9073 |
| VideoTetris (Ours) | **0.5563** | **0.2350** | **0.9312** |

## A.4 User Study

To verify the effectiveness of our proposed VideoTetris, we conduct an extensive user study across various scenes and models. Users compared model pairs by selecting their preferred video from three options: method 1, method 2, and comparable results. As presented in fig. 7, our method (orange in left) obtains more user preferences than others (blue in right), which further proving its effectiveness.

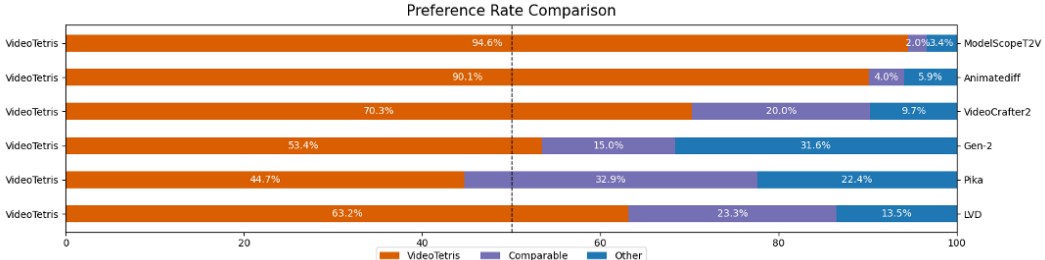

Figure 7: User Study about Comparision with Other Video Generation Methods

## A.5 Model Hyperparameters and Implementation Details

In this section, we further detailed our model hyperparameters and our implementation details. The hyperparameters in section 3.2 and section 3.3 are shown in table 8. In training process, we randomly drop out 5% of text prompts for classifier-free guidance training. We trained our model with batch size = 2 and learning rate = 1e-5 on 4 A800 GPUs for 16k steps in total.

## A.6 More examples

We provided more examples in the figures below.

Table 8: Hyperparameters of VideoTetris

| Dynamic-Aware Data Filtering | |
|---|---|
| $n_0$ | 4 |
| optical flow score threshold $s_1$ | 0.25 |
| optical flow score threshold $s_2$ | 0.75 |
| **Diffusion Training** | |
| Parametrization | $\epsilon$ |
| Diffusion steps | 1000 |
| Noise scheduler | Linear |
| $\beta_0$ | 0.0085 |
| $\beta_T$ | 0.0120 |
| Sampler | DDIM |
| Steps | 50 |
| $\eta$ | 1.0 |
| **Reference Frame Attention** | |
| 2D Conv input dim | 4 |
| 2D Conv output dim | 320 |
| 2D Conv kernel size | 3 |
| 2D Conv padding | 1 |
| MLP hidden layers | 1 |
| MLP inner dim | 320 |
| MLP output dim | 1024 |
| $l$ | 2 |

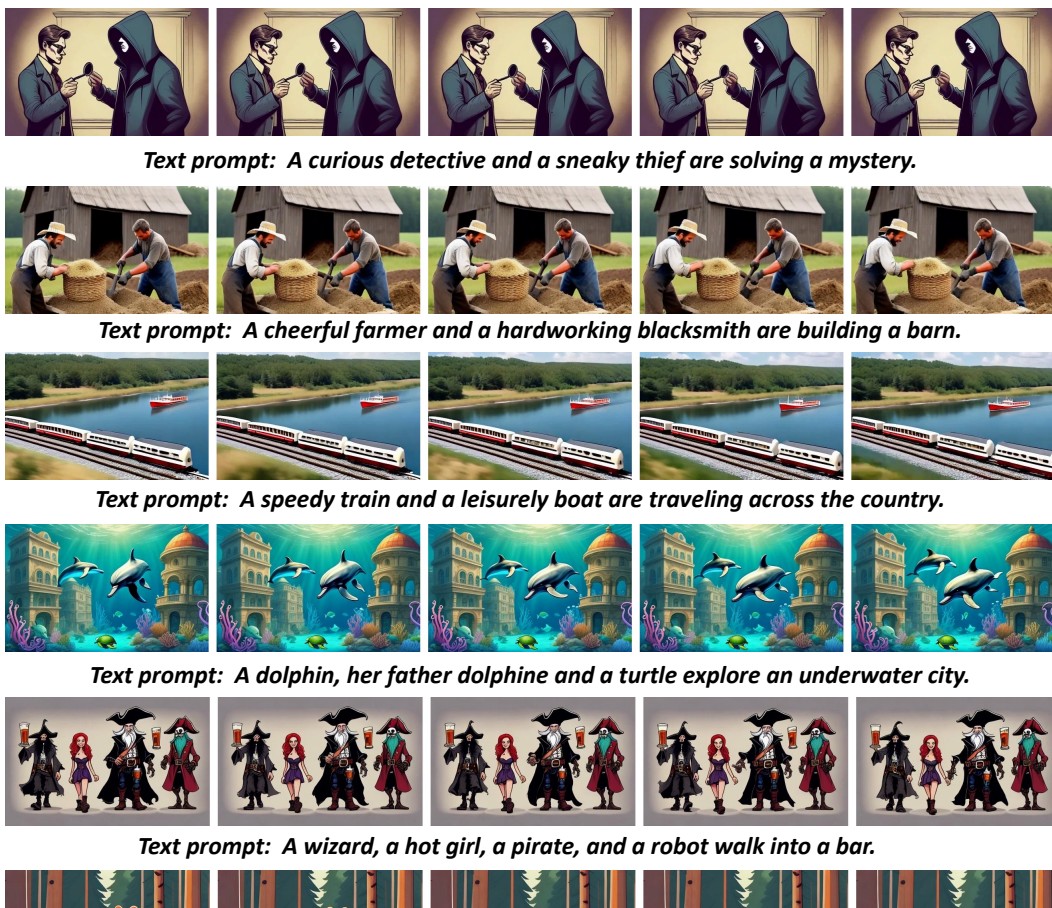

*Text prompt: A curious detective and a sneaky thief are solving a mystery.*

*Text prompt: A cheerful farmer and a hardworking blacksmith are building a barn.*

*Text prompt: A speedy train and a leisurely boat are traveling across the country.*

*Text prompt: A dolphin, her father dolphine and a turtle explore an underwater city.*

*Text prompt: A wizard, a hot girl, a pirate, and a robot walk into a bar.*

*Text prompt: A mother fox, a baby fox and a father fox go on a camping trip.*

Figure 8: More qualitative results of VideoTetris.

