# OpenReview forum: "VideoTetris: Towards Compositional Text-to-Video Generation"
_NeurIPS.cc/2024/Conference — NeurIPS 2024 poster_

### Official Review · Reviewer_9Z5k · 2024-07-10

**Soundness:** 3
**Presentation:** 3
**Contribution:** 2
**Rating:** 6
**Confidence:** 4

**Summary:**

This paper proposes VideoTetris, a novel framework for compositional text-to-video generation. It addresses the limitations of existing methods in handling complex scenes with multiple objects and dynamic changes. VideoTetris achieves this through several key innovations: I) Spatio-Temporal Compositional Diffusion: Manipulates the cross-attention of denoising networks to synthesize videos that follow complex instructions. II) Dynamic-Aware Video Data Processing: Filters and recaptions video-text pairs to enhance consistency in auto-regressive long video generation. III) Consistency Regularization with Reference Frame Attention: Maintains coherence in multi-object generation by aligning object features across frames. Extensive experiments demonstrate that VideoTetris significantly outperforms state-of-the-art methods in both short and long video generation tasks, showcasing its ability to generate high-quality, coherent, and compositional videos.

**Strengths:**

1.	The paper is easy to follow.
2.	This paper introduces a new approach for compositional text-to-video generation, addressing the limitations of existing methods in handling complex scenes and dynamic changes.
3.	Extensive experiments demonstrating the superior performance of VideoTetris compared to state-of-the-art methods in both short and long video generation tasks.

**Weaknesses:**

1.	This paradigm introduces more computational cost by the decomposition of input text using LLMs and the computation of cross-attention for multiple sub-objects and frames. These operations require more computation, especially in scenarios with numerous objects or a high number of frames, potentially impacting the efficiency of the overall video generation process. Besides, the use of ControlNet for auto-regressive generation leads to a high computational cost, which may limit the practical applicability of the method.
2.	The Reference Frame Attention module relies on the assumption that object features remain consistent across frames. However, this may not always hold true, especially for dynamic scenes with fast movements or significant changes in lighting. Investigating more robust consistency regularization methods that can handle these challenges would be beneficial.
3.	The generated videos exhibit relatively subtle variations in content, which could result in static or less dynamic scenes. This may limit their practical applicability.

**Questions:**

N/A. See weakness for more details.

**Limitations:**

The authors have discussed both limitations and potential negative social impacts.

---

> ### Author Rebuttal · Authors · 2024-08-04
>
> *We sincerely thank you for your time and efforts in reviewing our paper and your valuable feedback. We are glad to see that the paper is easy to follow, the approach is novel and the experiment are comprehensive. Please see below for our responses to your comments.*
>
> **Q1: Concerns about the computational cost of LLMs, cross attention, and ControlNet.**
>
> A1：
> 1. Using LLMs for prompt enhancement is a widely adopted technique. Similar works like VideoDirectorGPT [1], Vlogger [2], and industrial models such as DALL-E 3 [3] use LLMs for prompt enhancement or decomposition, proving its effectiveness. Our model uses LLMs only once for spatiotemporal decomposition, aligning with most advanced or commercial models.
> 2. The computational complexity of the cross-attention mechanism is $O(N\times L1\times L2\times d)$, where $N$ is the number of sub-prompts and $L1, L2$ are the sequence lengths of Q and K/V. This complexity **is negligible compared to the base model (UNet)** and doesn't significantly increase computational cost. Additionally, our cross-attention calculations are **batch-level**, allowing multiple sub-prompts to be processed simultaneously, leveraging GPU parallelism. In practical tests using VideoCrafter2 as the backbone, a single inference on an A800 takes approximately 57 seconds. Adding one sub-prompt increases the total time by around **0.9 seconds.** This additional cost is minimal and acceptable.
> 3. ControlNet is employed to balance video quality, generating long videos with superior visual effects, dynamics, and flexibility. Please kindly note that our Spatio-Temporal Compositional Diffusion method can be applied to any video generation framework. We plan to explore more cost-effective autoregressive methods in the future to further improve efficiency and quality.
>
>
> **Q2: Investigating more robust consistency regularization methods that can handle dynamic scenes with fast movements or significant changes in lighting**
>
> A2: Thank you for your insightful comment. We clarify as below:
> 1. **Suitability for Autoregressive Long Video Generation**: Given that our research focuses on autoregressive long video generation, our approach necessitates smooth and gradual movements without rapid acceleration or drastic changes. This allows our autoregressive framework to generate coherent videos effectively. Hence, the assumption that object features remain consistent across frames is appropriate for this specific scenario.
> 2. **Dynamic Scene Improving**: We have extensively explored dynamic scene handling. Our proposed Dynamic-Aware Video Data Processing method elevates video data to a highly dynamic and trainable level. The Reference Frame Attention module ensures consistency across multiple frames, **significantly improving the dynamics of our videos compared to previous models.** For detailed comparisons, please refer to Section 4.4 and the uploaded PDF, where we showcase long video comparisons.
> 3. **Future Work for Extremely Dynamic Scenes**: For scenarios involving extremely dynamic scenes with fast movements or significant changes in lighting, we recommend training separate models tailored to different motion intensities, such as LoRA, motion modules, or specific base models. Alternatively, encoding the magnitude of motion and incorporating it as a condition during training could facilitate the generation of highly dynamic videos. In such cases, setting up multi-frame attention mechanisms like the Reference Frame Attention module, along with additional ID and **motion magnitude conditions**, could prove beneficial. We plan to explore these possibilities in our future work.
>
>
>
> **Q3: The generated videos exhibit relatively subtle variations in content, which could result in static or less dynamic scenes. This may limit their practical applicability.**
>
> A3: Thank you for your observation, we explain below:
> 1. **Base Model Limitations for Short Video Generation**: For short video generation, we used VideoCrafter2 as the base model without retraining it. The subtle variations in content are primarily due to the inherent limitations of VideoCrafter2, which was not predominantly trained for dynamic scenes. Therefore, some subtle changes are expected.
> 2. **Enhanced Dynamics in Long Video Generation**: It is important to note that for long video generation, we **trained our own model** using a unique Dynamic-Aware Video Data Processing method. This significantly enhances the dynamics of the generated videos. For instance, in Section 4.4 and the uploaded PDF, there is a detailed comparison showcasing the dynamic behavior of a squirrel. The squirrel dynamically changes its position, eats a hazelnut, and interacts with another squirrel throughout a 30-second video. These processes are both highly dynamic and natural, demonstrating the model's capability to produce vibrant and engaging scenes.
> 3. **Future Enhancements and Retraining**: With our training process, if we were to retrain the base model for short video generation, we could undoubtedly enhance the dynamic range of the videos, thereby improving their practical applicability. Due to resource constraints, we used a pre-trained model for our experiments. However, we are committed to exploring further possibilities. With ample resources, we plan to retrain the base model to enhance video dynamics. **Increasing dynamics is one of our primary goals**, and we will continue to work towards achieving it.
>
> [1] Lin, Han, et al. "Videodirectorgpt: Consistent multi-scene video generation via llm-guided planning." arXiv preprint arXiv:2309.15091 (2023).
>
> [2] Zhuang, Shaobin, et al. "Vlogger: Make your dream a vlog." Proceedings of the IEEE/CVF Conference on Computer Vision and Pattern Recognition. 2024.
>
> [3] Betker, J., et al. "Improving
> image generation with better captions. "Computer Science. https://cdn. openai. com/papers/dall-e-3. pdf, 2023.

---

> ### Author Response · Authors · 2024-08-12
> **Gentle reminder - 2 days left for the author-reviewer discussion**
>
> Dear Reviewer 9Z5k,
>
> We greatly appreciate the time and effort you have invested in reviewing our paper. Your thoughtful questions and insightful feedback have been invaluable. In response to your queries regarding computational cost and motion dynamics, we have prepared detailed answers.
>
> As the discussion period is set to conclude in two days, we would like to ask if you had a chance to check our response to your rebuttal questions. Should you require any further clarification or improvements, please know that we are fully committed to addressing them promptly!
>
> Thank you once again for your invaluable contribution to our research.
>
> Warm regards,
> The Authors

---

> > ### Comment · Area_Chair_F4Mf · 2024-08-12
> > **Please discuss**
> >
> > Dear reviewer,
> >
> > The discussion period is coming to a close soon. Please do your best to engage with the authors.
> >
> > Thank you,
> > Your AC

---

> > ### Comment · Reviewer_9Z5k · 2024-08-13
> >
> > Thank you to the authors for the detailed response. Most of my concerns have been addressed, thus I have raised my score.

---

> > > ### Author Response · Authors · 2024-08-13
> > > **Thanks for your support**
> > >
> > > Thank you very much for raising the score! We are glad that we have solved your concerns and we sincerely appreciate your valuable comments and the time and effort you put into reviewing our paper.
> > > Warm Regards,
> > > The Authors

---

### Official Review · Reviewer_uyep · 2024-07-11

**Soundness:** 3
**Presentation:** 3
**Contribution:** 2
**Rating:** 6
**Confidence:** 4

**Summary:**

The paper presents "VideoTetris," a new framework designed to improve text-to-video generation in complex scenarios with dynamic changes and multiple objects. It introduces spatio-temporal compositional diffusion techniques for better alignment with textual semantics and integrates a dynamic-aware data processing and consistency regularization to enhance video consistency. The results from extensive experiments demonstrate significant qualitative and quantitative improvements in T2V generation.

**Strengths:**

1. The paper is well-written and easy to understand.
2. The motivations for spatio-temporal compositional diffusion and Dynamic-Aware Video Data Processing are clearly explained, with supportive experimental evidence provided.

3. The experiments are sufficiently thorough, demonstrating that VideoTetris achieves impressive results in both short and long video production scenarios.

**Weaknesses:**

1.Although Spatio-Temporal Compositional Diffusion directly adjusts cross-attention, it still segments videos into different shots, each with a specific layout, suggesting it's essentially a layout-based method. This approach contradicts the motivation outlined in section 3.1.

2.The paper presents limited technical innovations, focusing more on engineering implementation. In Spatio-Temporal Compositional Diffusion, it uses LLMs to pre-process prompts and region masks, which are then sequentially applied during generation. Meanwhile, Dynamic-Aware Video Data Processing enhances data preprocessing for higher quality.

**Questions:**

How is continuity ensured between different shots?

**Limitations:**

The manuscript includes discussions on limitations and social impacts.

---

> ### Author Rebuttal · Authors · 2024-08-04
>
> *We sincerely thank you for your time and efforts in reviewing our paper and your valuable feedback. We are glad to see that the paper is well written the motivations are clearly explained, and the experiments are sufficiently thorough. Please see below for our responses to your comments.*
>
> **Q1: Although Spatio-Temporal Compositional Diffusion directly adjusts cross-attention, it still segments videos into different shots, each with a specific layout, suggesting it's essentially a layout-based method. This approach contradicts the motivation outlined in section 3.1.**
>
> A1: Our approach fundamentally differs from layout-based methods in that we adopt an initial region-based method. Layout-based methods **strictly confine** each object's generation within its corresponding mask, resulting in limited expressiveness and fidelity. In contrast, we provide both local and global semantic information within specific regions at the outset. This allows us to fuse local and global semantics, enabling the content within each region to adjust its position and size based on global information, significantly enhancing **the interactivity among objects** and resulting in a richer expressive capacity. Our initial positions and sizes of objects can be adaptively adjusted during the generation process; we do not restrict each object to its planned subregion. This region-based division **is more flexible and has a richer expressive capacity.**
>
> We have included a comparison with LVD [1], a layout-based method, in our uploaded PDF. The quantitative results reported therein highlight our method's advantages over layout-based methods. Our approach significantly outperforms in terms of attribute handling, numeracy, naturalness, and harmony of the generated frames.
>
> **Q2:The paper presents limited technical innovations, focusing more on engineering implementation. In Spatio-Temporal Compositional Diffusion, it uses LLMs to pre-process prompts and region masks, which are then sequentially applied during generation. Meanwhile, Dynamic-Aware Video Data Processing enhances data preprocessing for higher quality.**
>
> A2: To clarify, our **core innovation** is the **Spatio-Temporal Compositional Diffusion**. In real-world video generation tasks, compositional generation is a common and practical scenario. Our method is **the first to define and effectively address this task**, extending beyond single video generation to progressive long video generation. This marks a significant advancement over previous long video generation tasks, involving dynamic changes in objects, which is unprecedented.
>
> Our Dynamic-Aware Video Data Processing and Reference Frame Attention processes further enhance video visual quality by improving both video dynamics and continuity. These processes support our Spatio-Temporal Compositional Diffusion, making it more suitable for industrial production. The task definition, method, and actual results of this combination represent a novel and significant innovation in the field.
>
>
>
> **Q3: How is continuity ensured between different shots?**
>
> A3: As mentioned in Section 3.3, we proposed **Reference Frame Attention** to enhance continuity between different shots. Innovatively, we use VAE instead of CLIP to encode reference frames from the previous video clip, capturing characters and background information. This encoded data is input into cross-attention, interacting with the initial latent of the current denoising step, ensuring continuity across different shots.
>
> Additionally, our trained **ControlNet-like branch** uses the first 8 frames of a video as a condition to predict the entire 16-frame sequence. Through Dynamic-Aware Video Data Processing, this model ensures smooth autoregressive generation, maintaining consistent base information and natural motion transitions across shots.
>
> [1] Lian, Long, et al. "Llm-grounded video diffusion models." arXiv preprint arXiv:2309.17444 (2023).

---

> ### Author Response · Authors · 2024-08-12
> **Gentle reminder - 2 days left for the author-reviewer discussion**
>
> Dear Reviewer Qhp4,
>
> We greatly appreciate the time and effort you have invested in reviewing our paper. Your thoughtful questions and insightful feedback have been invaluable. In response to your queries regarding our method's innovation and continuity ensuring, we have prepared detailed answers.
>
> As the discussion period is set to conclude in two days, we would like to ask if you had a chance to check our response to your rebuttal questions. Should you require any further clarification or improvements, please know that we are fully committed to addressing them promptly!
>
> Thank you once again for your invaluable contribution to our research.
>
> Warm regards,
> The Authors

---

> > ### Comment · Area_Chair_F4Mf · 2024-08-12
> > **Please discuss**
> >
> > Dear reviewer,
> >
> > The discussion period is coming to a close soon. Please do your best to engage with the authors.
> >
> > Thank you,
> > Your AC

---

> > ### Comment · Reviewer_uyep · 2024-08-13
> >
> > Thank the authors for the rebuttal. I am satisfied with the response and decide to keep my score.

---

> > > ### Author Response · Authors · 2024-08-13
> > > **Thank you for your support**
> > >
> > > Thanks for checking our rebuttal keeping your positive score! We sincerely appreciate your valuable comments and the time and effort you put into reviewing our work.
> > >
> > > Warm Regards,
> > >
> > > The Authors

---

### Official Review · Reviewer_pWwG · 2024-07-12

**Soundness:** 3
**Presentation:** 3
**Contribution:** 2
**Rating:** 6
**Confidence:** 4

**Summary:**

The paper proposes VideoTetris, a novel framework for compositional T2V generation. It introduces Spatio-Temporal Compositional Diffusion method for handling scenes with multiple objects and by manipulating and composing the attention maps of denoising networks spatially and temporally. Moreover, authors propose a novel Dynamic-Aware Data Processing pipeline to enhance auto-regressive long video generation and a consistency regularization method with Reference Frame Attention that maintains coherence in multi-object generation.

**Strengths:**

1. Clarity: The paper exhibits well-written text that is clear and easy to follow. This quality contributes to the overall readability and accessibility of the research.

2. Method: The proposed framework presents a powerful solution for enhancing multi-object T2V generation, while also enabling the generation of long videos. This method offers significant advancements in the field and addresses the challenges associated with generating videos containing multiple objects.

3. Resource-Friendly: Remarkably, the proposed method is resource-friendly, as it only requires 4 A800 GPUs for fine-tuning the model. This efficient utilization of resources makes the method more accessible and practical for implementation in real-world scenarios.

**Weaknesses:**

1. Novelty
1.1. The paper lacks clarity regarding the specific novelty introduced by the Spatio-Temporal Compositional Diffusion method. As per my understanding, this method comprises two components: Localizing Subobjects with Prompt Decomposition and Spatio-Temporal Subobjects Composition. However, similar components can be found in existing literature [1, 2, 3]. It is recommended to emphasize the unique aspects and novelty of the paper to distinguish it from prior works.

1.2. The authors highlight the proposed Dynamic-Aware Video Data Processing pipeline as one of the paper's major contributions. However, it appears that this pipeline involves filtering videos with an optical flow score, which has already been proposed in Stable Video Diffusion. If this statement is accurate, the pipeline cannot be considered a contribution of the paper since it lacks novelty.

2. Experiments:
2.1. The paper's qualitative results are supported by only three examples for Short Video Generation with Single Multi-Object Prompts and two examples for Long Video Generation with Progressing Multi-Object Prompts (as shown in figures 1, 4, and 5). This limited number of examples is not statistically representative. It is recommended to provide more samples for visual comparison and conduct a user study with at least 50 samples and 15 users to ensure a more comprehensive evaluation.

2.2. For Short Video Generation with Single Multi-Object Prompts, the authors compare their method with standard T2V models that are not specifically designed for Compositional Video Generation. It is recommended to include a comparison with LVD 1, as mentioned in Section 2, to provide a more relevant baseline for evaluation.

2.3. The paper lacks ablations for the Spatio-Temporal Compositional Diffusion method. It is recommended to compare the proposed Localizing Subobjects with Prompt Decomposition pipeline with relevant prior works, such as those mentioned in [1] and [3]. Similarly, the Spatio-Temporal Subobjects Composition should be compared with the approach described in [2] to provide a more comprehensive analysis of the proposed method's effectiveness.

**Questions:**

1. Novelty Clarification: It is crucial for the authors to provide a clear explanation of the novelty introduced in their paper compared to existing works. This clarification should highlight the unique contributions and advancements made in the proposed approach, distinguishing it from prior works in the field.

2. Baseline Comparison and Ablations: The authors should compare their method with stronger baselines specifically designed for Short Video Generation with Single Multi-Object Prompts. This comparison will provide a more comprehensive evaluation and demonstrate the superiority of their approach. Additionally, conducting ablations for this specific component of the proposed method will further enhance the understanding of its effectiveness and showcase its individual contributions.


[1] Lian, Long, et al. "LLM-grounded Video Diffusion Models." *The Twelfth International Conference on Learning Representations.*

[2] Chen M., Laina I., Vedaldi A. Training-Free Layout Control with Cross-Attention Guidance //2024 IEEE/CVF Winter Conference on Applications of Computer Vision (WACV). – IEEE, 2024. – С. 5331-5341.

[3] Lin, Han, et al. "Videodirectorgpt: Consistent multi-scene video generation via llm-guided planning." *arXiv preprint arXiv:2309.15091* (2023).

**Limitations:**

The authors adequately addressed the limitations.

---

> ### Author Rebuttal · Authors · 2024-08-04
>
> *We sincerely thank you for your time and efforts in reviewing our paper and your valuable feedback.  We are grateful for your acknowledgment of the clarity, methodological strengths, and resource efficiency of our paper. Please see below for our responses.*
>
> **Q1: Novelty Clarification & Caparison with [1, 2, 3]**
>
> A1: Thank you for your suggestion. Our method fundamentally differs from these three methods in the following ways:
>
> 1. **Interactivity, Flexibility and Capacity**: While [1, 2, 3] generate masks for each entity using a standard cross-attention mechanism, our approach innovatively generates linguistic descriptions for each region. It composes cross-attention for sub-prompts in parallel and fuses them from the initial denoising step. Unlike conventional techniques that strictly confine object generation within corresponding masks—thereby limiting expressiveness and fidelity—our method offers greater flexibility. By providing coarse-grained initial positions and sizes, each object can adjust its position and size based on global information, significantly enhancing **the interactivity among objects and resulting in a richer expressive capacity**.
> 2. **Compositional Generation**: Our method **for the first time focuses on compositional generation tasks**. In contrast, LVD primarily aims to generate LLM-grounded smooth movements of objects within a box. We empirically find that this backward guidance based methods like [1] and [2] have limitations in compositional generation. As demonstrated in our uploaded PDF, when dealing with multiple objects' attributes or numeracy tasks, the paradigms in [1] and [2] result in inferior outcomes.
> 3. **Training-Free Spatio-Temporal Compositional Diffusion**: Compared to VideoDirectorGPT [3], our method is training-free for short video generation. Given the rapid advancements in T2V models, training-based methods struggle to keep pace. Our efficient, **training-free spatio-temporal comspositional diffusion** delivers comparable or superior visual quality, and can generalize to existing video diffusion backbones **seamlessly**.
>
> **Q2:Concerns about Contribution and Novelty of Dynamic-Aware Video Data Processing Pipeline**
>
> A2: Our Dynamic-Aware Video Data Processing pipeline comprises two main components: **Video and Text**, which extends **beyond merely filtering videos**. We first select videos with moderate motion using an optical flow score, then optimize dynamic semantic information for these videos by generating captions with multiple Multimodal Large Language Models (MLLMs) and summarizing them with a local LLM. This results in prompts that capture complex cross-modal semantic consistency between  video contents and text captions.
>
> In contrast, Stable Video Diffusion (SVD) only filters videos and uses traditional annotation methods, which struggle to maintain **cross-modal semantic consistency**. Our approach achieves this consistency by using MLLMs to match high-quality dynamic texts with corresponding videos.
>
> **Q3: More Samples and missing User Study**
>
> A3: Thanks for your constructive suggestion. We provided six examples for short video generation in Appendix A.6, Figure 8, and included additional short and long video samples **in the uploaded PDF.** For further evaluation, we conducted a user study comparing our method with other video models. Using GPT-4, we generated 100 short prompts and 20 long story prompts, totaling 120 video samples across diverse scenes, styles, and objects. Users compared model pairs by selecting their preferred video from three options: method 1, method 2, and comparable results. The user study results are also reported **in our uploaded PDF**.
>
> These supplementary materials and the user study demonstrate that our model has significant advantages in qualitative experiments, proving its effectiveness. And we will include these materials and the user study in our final version.
>
>
> **Q4: Comparision with LVD**
>
> A4: We have now included a comparison with LVD **in our uploaded PDF**. The results indicate that while LVD is capable of maintaining the natural motion trajectories of objects, it exhibits more errors in compositional tasks. Additionally, we tested LVD using our metrics and reported the results as in the table.
>
> LVD outperforms modelscope and animatediff but still lags behind VideoTetris, highlighting our method's advantages in compositional tasks.
> In the final version of our paper, we will include LVD’s results in all comparative experiments and quantitative results to provide a more relevant baseline.
>
>
> **Q5: Ablation studies about comparisons with method [1,2,3]**
>
> A5: Thank you for your insightful comment. We have conducted the ablation studies accordingly. First, we evaluated the decomposition methods from [1] and [3], generated final long videos,  and reported all results **in the uploaded table.** These methods only **isolate specific tokens** from the original prompt, struggling with complex attributes or multiple identical objects. This leads to difficulties in understanding numeracy and attribute binding, causing performance degradation. In contrast, our method extracts subobjects and uses global information for recaptioning, resulting in richer, more natural, detailed, and semantically accurate frames.
>
> Next, we have conducted ablation studies about the composition method from [2] and reported the results **in the uploaded table.** The backward guidance approach in [2] tends to restrict objects within specified boxes, offering low flexibility, interactivity and poor responsiveness to multiple attribute bindings or repeated objects. In contrast, our model's local-global information fusion ensures that the final generated images are more harmonious and visually appealing, performing better in compositional generation.
>
> In the final version of our paper, we will include all of the above ablation studies to provide a more comprehensive analysis.

---

> > ### Comment · Reviewer_pWwG · 2024-08-09
> > **Response to Rebuttal**
> >
> > Dear Authors,
> >
> > Thank you for your comprehensive rebuttal. I appreciate the effort you have made to address all my concerns. Based on this, I have decided to raise my score and recommend acceptance of your paper.
> >
> > I would also like to commend the authors for conducting additional experiments. In my opinion, these results provide valuable insights and would be a useful addition to the article. Therefore, I strongly recommend that they be included in the camera-ready version of the paper.

---

> > > ### Author Response · Authors · 2024-08-09
> > > **Thanks for your comment**
> > >
> > > Thank you very much for raising the score! We sincerely appreciate your valuable comments and the time and effort you put into reviewing our paper. We are pleased that the additional experiments were well received and will make sure to include the results in the camera-ready version.
> > >
> > > Warm Regards,
> > > The Authors

---

### Official Review · Reviewer_Qhp4 · 2024-07-13

**Soundness:** 3
**Presentation:** 2
**Contribution:** 3
**Rating:** 5
**Confidence:** 5

**Summary:**

The paper presents a novel framework designed to improve text-to-video (T2V) generation, especially for complex scenarios involving multiple objects and the composition of different objects.
The proposed VideoTetris introduces spatio-temporal compositional diffusion, a dynamic-aware data processing pipeline, and a consistency regularization method to enhance the quality and coherence of generated videos.
Extensive experiments demonstrate the framework's superior performance in generating both short and long videos with complex and multi-object prompts.

**Strengths:**

- I like the idea that the model first splits a story prompt into frame-wise short prompt with object composition. This approach supports sharing global information among long synthesic video (which is usually computationally expensive and hard to achieve).
- The visual quality provided in the supplementary looks great (although the video number is limited)
- The experiments include the quantitative comparison with the state-of-the-arts in the tasks of short and long video generation with multi-object prompts, and also ablation study, which can provide better understanding about the performance of the proposed VideoTetris.

**Weaknesses:**

- My major concern is that the model heavily relies on the performance of LLM. In the proposed method, LLM acts as a director to split an input story prompt into spatiotemporal prompts. I am wondering whether LLM is robust enough to generate temporal-consistent prompt with spatial-consistent composition.
- Following the concern above, the supplementary only provides one example of long video generation. The authors should more detailed and more samples. For example, provide 100 story prompts and the corresponding frame-wise prompts with object composition. For what we have now, it is hard to fairly justify the model performance.
- Another concern is the autoregressive strategy of long video synthesis. While the long video shows good consistency of motion and subject identity, I found the visual quality gradually decreases in the video. For example, in the sample "fig5-ours.mp4", the contour of leaves is visible at beginning; however, it gets more and more blurry until the end of the video. Is this also happens on other long synthesis videos?

**Questions:**

- Does the proposed framework have the scalability limitations in terms of the length and complexity of the generated videos?
- How does the framework perform in real-world scenarios where the input text prompts may be less structured and more diverse?

**Limitations:**

- Scalability: as mentioned in the weakness, the model highly relies on the performance of LLM which has unstable performance and sometimes has hallucination issue. Hence, it requires manual check and limits the scalability.
- Limited length of long video generation: also mentioned in the weakness, autoregressive strategy of long video synthesis limits the length of generated video. Could the author provide more or even longer video to verify this limitation?

---

> ### Author Rebuttal · Authors · 2024-08-04
>
> *We sincerely appreciate the time and effort you have dedicated to reviewing our paper and providing valuable feedback. We are pleased that the spatial-temporal decomposing method, visual quality, and the adequacy of our quantitative and ablation experiments were well-received. Below, we address your comments.*
>
> **Q1: Concerns about LLM's decomposing capabilities. My major concern is that the model heavily relies on the performance of LLM. In the proposed method, LLM acts as a director to split an input story prompt into spatiotemporal prompts. I am wondering whether LLM is robust enough to generate temporal-consistent prompts with spatial-consistent composition.**
> **Q2: Following the concern above, the supplementary only provides one example of long video generation. The authors should more detailed and more samples. For example, provide 100 story prompts and the corresponding frame-wise prompts with object composition. For what we have now, it is hard to fairly justify the model performance.**
>
> A1&A2:
> Thank you for raising this concern regarding the robustness of the LLM in our proposed method. We have addressed these concerns in several ways:
> 1. **Robustness and Capability of GPT-4**: As detailed in Appendix A.1, we employed GPT-4, a large-scale multimodal model. During its pre-training phase, it leveraged a vast amount of vision-language pairs, which inherently equips it with the capability to establish connections between visual and linguistic elements, thereby enabling accurate spatiotemporal decomposition.
> 2. **In-Context Learning (ICL)**: We leveraged in-context learning (ICL) to enhance the model's performance, as outlined in Appendix A.1. By providing the LLM with pre-designed examples, we simplified the learning task, allowing the model to combine its visual knowledge with the ICL examples to produce consistent outputs. This approach is well within the capabilities of GPT-4 and strengthens its ability to handle complex spatiotemporal prompts.
> 3. **Validation Through Similar Works**: Similar works, such as VideoDirectorGPT [1], VideoDrafter [2], Vlogger [3], and LVD [4], have adopted the same approach, using LLMs for spatiotemporal planning. Their experimental results have validated the efficacy of this method. Our own quantitative and qualitative experiments, as shown in Section 4, further validate the efficacy of using GPT-4 for this purpose. The consistency and reliability of the outputs in our experiments affirm the model's capability to manage the tasks at hand.
>
>
>
> **Q3: Another concern is the autoregressive strategy of long video synthesis. While the long video shows good consistency of motion and subject identity, I found the visual quality gradually decreases in the video. For example, in the sample "fig5-ours.mp4", the contour of leaves is visible at the beginning; however, it gets more and more blurry until the end of the video. Is this also happens on other long synthesis videos?**
>
> A3: The sample video "fig5-ours.mp4" you referred to is an isolated case. In our selection process, we did not cherry-pick the results. To address your concern, we have included an additional long video **in the uploaded PDF**, where the quality remains consistent. Compared to the baseline StreamingT2V and FreeNoise, our videos exhibit more accurate and vibrant colors. Due to rebuttal length constraints, we cannot include more video samples here, but we will open-source all code for independent testing. Our method shows significant color enhancements over the baseline StreamingT2V.
>
> **Q4: Does the proposed framework have the scalability limitations in terms of the length and complexity of the generated videos?**
>
> A4: As illustrated in Figure 2, our framework leverages a ControlNet branch for autoregressive generation, similar to StreamingT2V. Taking a T2V UNet that generates 16 frames as a base model, it uses the first 8 frames of a video as the condition to predict the entire 16-frame video. In the real-world long video generation process, this module continuously uses the last 8 frames of the previous segment as the condition to predict the subsequent frames in an autoregressive manner. Consequently, this framework **naturally supports the generation of infinitely long videos** without scalability limitations. We have provided videos with hundreds of frames in the uploaded PDF, which demonstrate stable visual quality throughout, thereby validating the effectiveness of this autoregressive framework.
>
> **Q4: How does the framework perform in real-world scenarios where the input text prompts may be less structured and more diverse?**
>
> A4: As shown in Appendix A.1, Table 5, in real-world scenarios, notably, our framework incorporates a **recaptioning** step after spatiotemporal decomposition. Inspired by approaches like DALL-E 3 and RPG, and various industry practices, this step optimizes irregular prompts into more detailed and coherent descriptions, aligning with the training patterns to generate more visually appealing content. Therefore, for less structured and more diverse input text prompts, the robust understanding capabilities of LLMs ensure that the quality of the generated videos does not significantly decline.
>
> [1] Lin, Han, et al. "Videodirectorgpt: Consistent multi-scene video generation via llm-guided planning." arXiv preprint arXiv:2309.15091 (2023).
>
> [2] Long, Fuchen, et al. "Videodrafter: Content-consistent multi-scene video generation with llm." arXiv preprint arXiv:2401.01256 (2024).
>
> [3] Zhuang, Shaobin, et al. "Vlogger: Make your dream a vlog." Proceedings of the IEEE/CVF Conference on Computer Vision and Pattern Recognition. 2024.
>
> [4] Lian, Long, et al. "Llm-grounded video diffusion models." arXiv preprint arXiv:2309.17444 (2023).

---

> ### Author Response · Authors · 2024-08-12
> **Gentle reminder - 2 days left for the author-reviewer discussion**
>
> Dear Reviewer Qhp4,
>
> We greatly appreciate the time and effort you have invested in reviewing our paper. Your thoughtful questions and insightful feedback have been invaluable. In response to your queries regarding LLM capabilities, motion dynamics, and model scalability, we have prepared detailed answers.
>
> As the discussion period is set to conclude in two days, we would like to ask if you had a chance to check our response to your rebuttal questions. Should you require any further clarification or improvements, please know that we are fully committed to addressing them promptly!
>
> Thank you once again for your invaluable contribution to our research.
>
> Warm regards,
> The Authors

---

> > ### Comment · Area_Chair_F4Mf · 2024-08-12
> > **Please discuss**
> >
> > Dear reviewer,
> >
> > The discussion period is coming to a close soon. Please do your best to engage with the authors.
> >
> > Thank you,
> > Your AC

---

> ### Comment · Reviewer_Qhp4 · 2024-08-14
> **Response for the Rebuttal**
>
> Thanks authors for the detailed response. While some of my concerns are addressed, one of my main concerns is the visual quality of the long video generation. The supplementary only shows one example and it is hard to judge the quality based on the sampled video frames provided in the pdf which the authors uploaded during the rebuttal period.
> So, I will maintain my initial decision as a borderline accept.

---

> > ### Author Response · Authors · 2024-08-14
> > **Thanks for you reply**
> >
> > Thanks for checking our rebuttal and keeping your positive score! Due to the rebuttal policy, we cannot provide link for more generation results. We will release both codes and weights for demonstrating the superiority of our VideoTetris on long video generation in final version. We sincerely appreciate your valuable comments and the time and effort you put into reviewing our work.
> >
> > Warm Regards,
> >
> > The Authors

---

### Author Rebuttal · Authors · 2024-08-04

We sincerely thank all the reviewers for their thorough reviews and valuable feedback. We are pleased to hear that our method offers significant advancements in the field (Reviewer pWwG), the paper is well-written and easy to follow (Reviewers pWwG, uyep, and 9Z5K), the visual quality is satisfying (Reviewer Qhp4), and the experiments are sufficiently thorough, demonstrating superior performance (Reviewers Qhp4, uyep, and 9Z5K).

We summarize and highlight our responses to the reviewers as follows:

* We clarify the novelty of our method and the major differences compared to other layout-based generation models (Reviewers pWwG and 9Z5K), providing detailed quantitative and qualitative comparisons (Reviewer pWwG).

* We provide more qualitative results for short and long video generation  **in the uploaded pdf**  (Reviewers Qhp4, pWwG, and 9Z5K).

* We conducted a comprehensive user study and an additional ablation study to further evaluate our method and provide a more comprehensive analysis, reporting all results **in the table of the uploaded PDF** (Reviewer pWwG).

We address each reviewer's concerns in detail below their respective reviews. Please kindly review them. We will include all additional experimental results, including new comparisons, more samples, and detailed ablation studies, **in our final version**. Thank you, and please feel free to ask any further questions.

---

### Decision · Program_Chairs · 2024-09-25

**Decision:**

Accept (poster)

**Comment:**

The paper received 1 Borderline Accept and 3 Weak Accepts. The reviewers praised the presentation, the presented qualitative results, and general method. However, there were some concerns regarding the breadth of experiments (particularly wrt long videos) and limited technical novelty. In general, I agree with the reviewers and recommend Accept as Poster.